# Trait emotion regulation predicts momentary self-esteem level and variability in adolescents' daily lives
Dennis Warnholtz ✉, Larissa Lucia Wieczorek, Eva Bleckmann & Jenny Wagner

Level and variability are two key aspects of momentary self-esteem that are associated with mental health from early adolescence onwards, but little is known about their determinants. The current study examines how the trait-level use of four emotion regulation strategies—reappraisal, reflection, expressive suppression, social sharing—is associated with the level and variability of momentary self-esteem in the developmentally critical period of adolescence. Using mixed-effects location scale models, we analyzed experience sampling data from 408 adolescents (14–22 years, 81.62% girls) who reported their momentary self-esteem up to 35 times across one week. Two findings stand out: first, adolescents who tended to engage more in reappraisal, reflection, and social sharing to regulate their emotions experienced higher momentary self-esteem levels, whereas those who tended to engage more in expressive suppression experienced lower levels. Second, the tendency to use expressive suppression was consistently linked to more variability in momentary self-esteem. We discuss the contribution of emotion regulation strategies to levels of momentary self-esteem in adolescence and highlight the need for further research into the mechanisms underlying its variability.

Being shaped by daily experiences such as social inclusion or encounters of success and failure, self-esteem—defined as a person's evaluation of their own value in specific situations—has been shown to fluctuate at the momentary level[1–3]. This momentary self-esteem is characterized by two key aspects: its level, describing how people feel about themselves on average across situations, and its variability, capturing how much people fluctuate in their self-evaluations across time and situations[4]. Both momentary self-esteem level and variability are important in relation to general functioning and mental health[5,6]. In particular, momentary self-esteem variability has been linked to different mental disorders[7–10]. However, previous research has primarily examined the outcomes associated with momentary self-esteem variability, while less attention has been given to its contributing factors. Moreover, most of the existing evidence stems from studies of undergraduates and adults, leaving a limited understanding of momentary self-esteem during adolescence.

Adolescence is a crucial phase for gaining insight into the dynamics of momentary self-esteem for at least two reasons. First, adolescence is a critical period for the development of self-esteem, as indicated by both relatively low mean-levels and rank-order stabilities of trait self-esteem, the stable component of self-esteem[11]. Second, momentary self-esteem is already associated with mental health outcomes at this sensitive age, which marks the onset of many mental disorders[12,13]. Since momentary self-esteem is essentially an emotional evaluation of oneself—either positively or negatively—a crucial factor in understanding self-esteem levels and variability in everyday life is how well adolescents can manage their emotions. Adolescence is often seen as a time of emotional turmoil[14]. Compared to children and adults, adolescents show more frequent, intense, and unstable emotional expressions[15,16]. Furthermore, adolescents experience a shift toward feeling fewer positive and more negative emotions as they transition from early to late adolescence[17]. These shifts in adolescents' emotional expressions highlight the importance of developing a more nuanced set of emotion regulation (ER) strategies, defined as strategies that people use to manage which emotions they experience, when they occur, and how they are expressed[18].

Based on the theoretical notions of the Process Model of ER[18,19], a wide range of ER strategies have been explored. The current project examines four strategies: reappraisal (re-evaluating the meaning or relevance of a potentially emotional situation to oneself[19]), reflection (engaging in self-attention through curiosity or a desire for self-knowledge[20,21]), expressive suppression (inhibition of one's own emotional expression[19]), and social sharing (communicating with others about the circumstances of an emotion-eliciting event and one's own emotional reactions[22]). Reappraisal and reflection are considered cognitive ER strategies that are implemented before or during the emergence of an emotion to shape the final emotional response[20,23,24]. By contrast, expressive suppression and social sharing are considered behavioral ER

Department of Educational Psychology and Personality Development, University of Hamburg, Hamburg, Germany. ✉e-mail: Dennis.Warnholtz@uni-hamburg.de

strategies that are used to manage emotional responses that have already unfolded[20,23,24].

The emotional, cognitive, and social changes during adolescence are pivotal for the development of ER, fostering the use of more cognitive, sophisticated strategies, and greater flexibility in their application[25–27]. Reappraisal emerges as an effective strategy protecting adolescents from negative mental health outcomes[25,26,28]. Although less is known about the use of reflection, adolescents become more capable of attending to and reflecting on their own emotions[25,27]. Expressive suppression is already employed by children[29], and its use as behavioral ER strategy increases through adolescence[30]. Expressive suppression is often associated with negative mental health outcomes, but it may be beneficial for psychological functioning when employed in a context-sensitive manner[29]. The frequency and role of social sharing in adolescence remain unclear. Notably, social sharing per se does not effectively reduce negative affect, as it can involve excessive problem discussion and dwelling on negative feelings, which is associated with negative mental health outcomes[31]. While we have discussed each ER strategy individually, it has generally been argued that it is most adaptive to have a broad repertoire of ER strategies in adolescence[32].

Focusing on the links between ER strategy use and self-esteem, existing research has largely focused on trait self-esteem and adult samples, suggesting that reappraisal relates to higher and expressive suppression to lower trait self-esteem[33]. In adolescence, reappraisal likely mitigates the negative effect of peer victimization on self-esteem[34]. Additionally, self-reflection tends to be associated positively with the self-esteem of adults with high self-esteem and negatively among those with low self-esteem[35]. Finally, support seeking through whining and expressing sadness without revealing the underlying cause—a potential facet of social sharing—is linked to lower self-esteem[36]. Conversely, sharing positive emotions is linked to higher-quality social relationships[37] and may also contribute to higher self-esteem via sociometer processes[38]. Together, previous research highlights the relevance of ER for individual differences in trait self-esteem of adults. However, it remains unclear how different ER strategies relate to momentary self-esteem fluctuations (i.e., level and variability) in daily life, especially in adolescence.

The present research investigates how trait ER strategy use relates to momentary self-esteem level and variability in adolescence. We addressed two key research questions: First, to what extent do the four ER strategies reappraisal, expressive suppression, social sharing, and reflection explain the level of adolescents' momentary self-esteem? Second, to what extent do ER strategies explain the variability of adolescents' momentary self-esteem? Based on trait ER and trait self-esteem associations in adult samples[33], we hypothesized that reappraisal is positively, and expressive suppression is negatively associated with momentary self-esteem levels. All remaining associations, that is, those of social sharing and reflection with momentary self-esteem levels, as well as those of all four ER strategies with momentary self-esteem variability, were tested exploratorily.

## Methods

The present study is a correlational study based on convenience sampling. We combined data from two original experience sampling (ESM) studies with adolescent participants. The SELFIE[39] study, from now on Study 1, includes data from students in their final year of high school ($M_{age}$ = 17.64, SD = 0.93); the SchoCo[40] study, from now on Study 2, includes data from students in different school tracks ($M_{age}$ = 15.89, SD = 1.22). The codebooks for Study 1 (https://osf.io/4gnz9/) and Study 2 (https://osf.io/r5gjx/) contain detailed information on the full list of items, response formats used, and the data collection process. All hypotheses and analyses of the present study were preregistered on February 2, 2025 (https://osf.io/wdxfe). Online Supplementary Materials (OSM) are provided on the OSF (https://osf.io/9ntck/).

## Participants

The combined sample comprised 461 adolescents, with 220 participants from Study 1 and 241 from Study 2. We excluded 53 participants because they reported fewer than three ESM reports, which are necessary to adequately model variability in momentary self-esteem. Thus, the final sample comprised 408 adolescents (81.62% female, 18.38% male) aged 14–22 years ($M_{age}$ = 16.83, SD = 1.41), who recorded a total of 8349 ESM entries, with an average of 20.49 ESM entries per person (SD = 9.91, range: 3–35). See Supplementary Tables 1 and 2 in the OSM for differences between excluded and included participants, and between participants of Study 1 and Study 2, respectively. Based on a simulation-based power analysis of a prior study that used the same data set and conducted similar types of analyses with different variables[16], we expected a satisfactory power well-above 0.80 to detect small effects with standardized $\beta$-coefficients equal to or larger than 0.10 at an alpha level of 0.05.

## Procedure

The data collection procedures for both samples were similar: adolescents completed several online questionnaires assessing demographic information and various self-report measures (e.g., ER) before entering a week-long ESM phase. During the ESM-phase participants received questionnaires on their own smartphones five times a day (9 a.m., 12 p.m., 3 p.m., 5 p.m., and 8 p.m.). Answering all items in the questionnaires was mandatory; however, participants could skip individual ESM measurements, leading to varying numbers of completed ESM reports across participants. The questionnaires were implemented with the open-source software *formr*[41]. Social media platforms, personal contact with schools and flyers in public spaces were used to promote both studies. In Study 1, adolescents were compensated financially in proportion to the number of completed questionnaires and ESM reports, up to a maximum of 150€. They also received personalized feedback and had the opportunity to win prizes upon completing the entire study. In Study 2, adolescents could participate in a lottery, in which sweets and three gift cards with a value of 50€, 75€, and 100€ were raffled. Further, they received personalized feedback upon full completion of the study. More information on participant compensation can be found in the respective codebooks. The German Psychological Society (DGPs, protocol code JW 052014rev; date of approval: 2014-08-20) and the ethics committee of the Department of Psychology at the University of Hamburg granted ethical approval for the Study 1 and Study 2 studies, respectively. For both studies, informed consent was obtained from all individual participants included in the study.

## Measures

Both studies used the same measures for the examined constructs. While momentary measures were recorded during the ESM week, all remaining variables were measured in the preceding online questionnaire.

*Momentary self-esteem* was assessed with a single item ("All things considered, how content are you with yourself right now?"), which was adapted from the Rosenberg Self-Esteem Scale[42]. Participants were asked to respond to the item on a 10-point rating scale from 1 (not at all) to 10 (very). We calculated the intraclass correlation coefficient (ICC) Type 2 to assess the reliability of an individual's average momentary self-esteem across ESM entries[43]. In the current sample, the ICC(2) indicated a high reliability of .96.

*ER strategies* were assessed using four subscales: Reappraisal (e.g., In general, I regulate my emotions by thinking that I can learn something from that situation.) and expressive suppression (e.g., In general, I regulate my emotions by not showing them to someone else.) were assessed with three items each. Reflection (e.g., In general, I regulate my emotions by thinking about what the situation caused/triggered) and social sharing (e.g., In general, I regulate my emotions by telling someone what has happened) were assessed with two items each. All ER strategy items were answered on a scale ranging from 1 (strongly disagree) to 10 (strongly agree) and summarized as average scores for each ER strategy. The reappraisal and reflection items were adapted from the Cognitive Emotion Regulation Questionnaire[44] and its short version[45]. Suppression items were derived from the Emotion Regulation Questionnaire[33] and social sharing items were adapted from Brans et al.[20]. To evaluate the internal consistency of three or more item scales, we report McDonald's omega ($\omega$)[46] for two item scales, we report the Spearman-Brown coefficient ($r_{SB}$)[47]. Internal consistencies were good across

scales ($\omega = 0.84$ for reappraisal, $\omega = 0.88$ for expressive suppression, $r_{SB} = 0.71$ for reflection, $r_{SB} = 0.89$ for social sharing).

*Control variables* were included at the occasion and the person level. At the occasion level, participants were first asked whether another person was present (0 = not present, 1 = present). Second, we coded whether the measurement took place during the weekend (0 = weekday, 1 = weekend). Finally, we computed a time variable, indicating the sequential order of the ESM assessments from 1 to 35, corresponding to the five assessments across 7 days. At the person level, participants reported their age and gender (0 = female, 1 = male) and we coded the original sample (0 = Study 1, 1 = Study 2). Participants self-reported their gender. We did not specify whether responses should reflect gender identity or biological sex, as the German term "Geschlecht" can refer to either. Gender was only considered in the robustness checks as reported in the method and results section. Trait self-esteem was assessed with a German short form of the Rosenberg Self-Esteem Scale[42,48,49], rated on a 7-point scale (1 = strongly disagree, 7 = strongly agree; $\omega = 0.93$).

## Data analyses
To address our research questions, we estimated a series of mixed-effects location scale models (MELSM)[50]. These models allow for the simultaneous estimation of both the level and variability of momentary self-esteem in a single model, while accounting for their potential correlation to avoid biased estimates[50,51]. All models were structured on two levels, with measurement occasions (level 1) nested within participants (level 2). For model computation, we used the Bayes estimator with Markov Chain Monte Carlo methods, a default of two chains, and Mplus' default diffuse priors[50,52]. A time variable was included in all models as a continuous index of ESM progress to control for potential developmental trends[53]. All continuous between-person predictors were centered on their respective grand means.

We fitted four separate models where adolescents' momentary self-esteem levels and variability were regressed on each of the four ER strategies, respectively. These models are written as:

Level 1:

$$Momentary\ SE_{ti} = \beta 0_i + \beta_{1i}\,time_t + r_{ti} \tag{1}$$

Level 2:

$$\begin{aligned}\beta_{0i} &= \gamma_{00} + \gamma_{01} reappraisal_i + u_{0i}\\ \beta_{1i} &= \gamma_{10} + u_{1i}\end{aligned} \tag{2}$$

where

$$r_{ti} \sim N(0, \sigma_i^2) \tag{3}$$

and

$$\sigma_i^2 = \exp[\omega_0 + \omega_1 reappraisal_i + u_{2i}] \tag{4}$$

At Level 1, momentary self-esteem at measurement $t$ of person $i$ is modeled as a function of time, with a person-specific intercept $\beta_{0i}$, a person-specific effect of time $\beta_{1i}$ and a residual $r_{ti}$ for each person and measurement. The variance of the within-person residuals is assumed to vary across individuals (Eq. 3).

At Level 2, between-person differences in the person-specific intercept $\beta_{0i}$ (i.e., the individual momentary self-esteem level) were modeled by regressing it on the self-reported general tendency to use one of four different ER strategies (here exemplary for reappraisal). The random slope $\beta_{1i}$ of the effect of time indicates that longitudinal trends of momentary self-esteem across the ESM week can vary across participants. In Eqs. 2 and 4, $u_{0i}$, $u_{1i}$, and $u_{2i}$ represent random location intercept, random location slope, and random scale intercept, respectively, which are assumed to follow a multivariate normal distribution[50].

Between-person differences in the within-person residual variance $\sigma_i^2$ (i.e., the individual momentary self-esteem variability; Eq. 4) were then modeled by regressing it on adolescents' self-reported tendency to use of ER strategies. Given that variances cannot be negative, the within-person residual variance was modeled within an exponential function as a log-linear model, which precludes negative values.

We tested the robustness of our results in two ways: First, we re-estimated our models with additional control variables at the within-person level (presence of an interaction partner, weekday vs. weekend) and at the between-person level (gender, age, and the sample of the original study). Second, we estimated a MELSM including all ER strategies simultaneously as predictors of momentary self-esteem level and variability.

In addition, we conducted two exploratory follow-up analyses: first, we tested whether the number of ER strategies a person used was associated with their momentary self-esteem level or variability. For each ER strategy we created two dummy-coded variables using different approaches: absolute coding, with individual means below 3 being coded as 0 (non-usage), and means of 3 or higher as 1 (usage); and relative coding, with individual means below one standard deviation of the sample mean being coded as 0 (non-usage), while those within one standard deviation of the sample mean and above being coded as 1 (usage). The threshold for absolute coding was chosen arbitrarily due to a lack of established cutoffs. The threshold for relative coding was based on Lougheed & Hollenstein[32]. Sum scores for each approach illustrate the use of zero to four ER strategies. Second, we estimated a set of MLSEMs with both ER strategies and trait self-esteem as predictors of momentary self-esteem to examine predictive effects beyond stable trait differences in self-esteem. This approach deviates from the preregistration in two ways: first, we specified trait self-esteem as a general control variable. Given the high correlation between state and trait self-esteem in previous studies[3], the inclusion of trait self-esteem should be considered a very conservative test. Thus, we estimate a separate exploratory model rather than including trait self-esteem in all models containing control variables. Second, an error in the preregistration suggested only momentary self-esteem variability would be regressed on trait self-esteem. In reality, we intended to estimate the more conservative model predicting both level and variability. Further details are provided in the Supplementary Note 1 in the OSM.

Convergence of all models was evaluated based on the potential scale reduction factor (PSR[54,55]) and the inspection of trace and autocorrelation plots. We specified 20,000 iterations per chain for all models, which included a burn-in period of 10,000 iterations. For all estimated models, a PSR of 1 indicated good convergence. Based on Bayesian estimation, we reported the medians and 95% credible intervals of the posterior distributions for the parameters of interest. Parameters whose 95% credible intervals did not include zero were considered statistically significant. Additionally, we computed $R^2$ as a measure of effect size, representing the proportion of explained variance relative to a null model[56].

## Reporting summary
Further information on research design is available in the Nature Portfolio Reporting Summary linked to this article.

## Results
Descriptive statistics and intercorrelations of the within and between-person variables are presented in Tables 1 and 2. The ICC(1) indicated that 54% of the variance in momentary self-esteem was explained by between-person differences, pointing to substantial within-adolescent variability alongside systematic between-person differences in momentary self-esteem. Figure 1 illustrates momentary self-esteem levels and variability for individuals with low, average, and high levels of variability. In the following, we focus on the models without control variables and report deviations from the models accounting for covariates. The full set of results for all MELSMs is available in the OSM provided on the OSF. The results of the robustness analysis are presented in Supplementary Tables 3 and 4, and the results of our exploratory analysis are presented in Supplementary Tables 5 and 6.

## Emotion regulation strategies predicting momentary self-esteem level and variability

The results of the MELSMs addressing the two research questions on the associations between each of the four ER strategies and adolescents' momentary self-esteem level and variability are shown in Table 3. Adolescents who indicated a stronger tendency to regulate their emotions through reappraisal, reflection, and social sharing reported higher levels of momentary self-esteem, whereas expressive suppression was associated with lower levels. In addition, reappraisal was associated with lower self-esteem variability and expressive suppression with higher momentary self-esteem variability. It should be noted that the estimates for momentary self-esteem variability are reported on a logarithmic scale. To interpret these estimates meaningfully, they should be converted back to the original scale by exponentiating the values. For example, a variability estimate of $-0.05$ for reappraisal corresponds to $\exp(-0.05) = 0.95$ on the outcome scale[50]. Thus, adolescents with a stronger tendency to use reappraisal exhibited lower self-esteem variability, whereas those who tended to engage in expressive suppression experienced greater fluctuations in their self-esteem. We found no statistically significant evidence that reflection or social sharing were associated with self-esteem variability. Across models, ER strategies accounted for a small to medium proportion of the variance in adolescents' momentary self-esteem level ($\Delta R^2_{Level} = 0.03-0.12$). In contrast, the models accounted for no or very small amounts of variance in momentary self-esteem variability ($\Delta R^2_{Variability} = 0.00-0.01$).

Our first robustness analysis (i.e., including control variables in all models; see Supplementary Table 3 in the OSM) illustrated that all effects of ER strategies on momentary self-esteem level remained robust (Reappraisal: $\gamma = 0.26$, 95% CI = [0.18, 0.33]; Expressive suppression: $\gamma = -0.24$, 95% CI = [$-0.30$, $-0.17$]; Reflection: $\gamma = 0.14$, 95% CI = [0.06, 0.22]; Social

sharing: $\gamma = 0.12$, 95% CI = [0.06, 0.19]). However, the effects on self-esteem variability were only partially robust: Specifically, the effect of expressive suppression on momentary self-esteem variability remained robust ($\omega = 0.04$, 95% CI = [0.00, 0.07]), but this was not true for reappraisal ($\omega = -0.04$, 95% CI = [$-0.08$, 0.01]). In addition, the covariates showed independent effects on momentary self-esteem level (estimates are reported from the models including reappraisal as ER strategy). At the within-person level, the presence of an interaction partner ($\gamma = 0.14$; 95% CI = [0.08, 0.21]) and weekends ($\gamma = 0.06$; 95% CI = [0.01, 0.12]) were associated with higher momentary self-esteem levels. At the between-person level, male adolescents reported higher momentary self-esteem than female adolescents ($\gamma = 0.64$; 95% CI = [0.19, 1.10]), while Study 2 participants showed lower momentary self-esteem levels compared to Study 1 participants ($\gamma = -0.58$; 95% CI = [$-1.02$, $-0.14$]). None of the covariates were significantly associated with momentary self-esteem variability (Age: $\omega = -0.04$; 95% CI = [$-0.12$, 0.05]; Gender: $\omega = -0.19$; 95% CI = [$-0.43$, 0.06]; Sample: $\omega = 0.12$; 95% CI = [$-0.13$, 0.36]). The ER strategies and covariates together explained small to medium proportions of variance in momentary self-esteem level ($\Delta R^2_{Level} = 0.09-0.15$) and variability ($\Delta R^2_{Variability} = 0.11-0.13$).

The second robustness analysis (i.e., including all ER strategies as predictors simultaneously; see Supplementary Table 4 in the OSM) yielded several additional insights: Whereas higher reappraisal ($\gamma = 0.22$; 95% CI = [0.14, 0.30]) and lower expressive suppression ($\gamma = -0.20$; 95% CI = [$-0.28$, $-0.12$]) were still linked to higher momentary self-esteem levels, the effects of reflection ($\gamma = 0.01$; 95% CI = [$-0.07$, 0.09]) and social sharing ($\gamma = -0.01$; 95% CI = [$-0.08$, 0.06]) were not significant anymore. As before, momentary self-esteem variability was positively predicted by expressive suppression ($\omega = 0.06$; 95% CI = [0.02, 0.10]) but not reappraisal ($\omega = -0.03$; 95% CI = [$-0.08$, 0.01]). In addition, the model considering all ER strategies simultaneously revealed a positive link between social sharing and variability ($\omega = 0.05$; 95% CI = [0.01, 0.09]). The ER strategies explained a moderate proportion of the variance in momentary self-esteem levels and a small proportion of the variance in its variability, as indicated by $\Delta R^2_{Level} = 0.18$ and $\Delta R^2_{Variability} = 0.03$, respectively.

## Exploratory analysis: the role of the number of ER strategies and trait self-esteem

To address the exploratory question whether the number of used ER strategies would affect momentary self-esteem level or variability, we estimated additional MELSMs with dummy-coded variables indicating the number of ER strategy use (see Supplementary Table 5 in the OSM). When using the absolute coding (individual means as indicators of (non)usage), the analyses provided no statistically significant evidence of an association between the frequency of used ER strategies and adolescents' momentary self-esteem levels ($\gamma = 0.27$; 95% CI = [0.00, 0.54]) and variabilities ($\omega = 0.06$; 95%

## Table 1 | Means, standard deviations, and intercorrelations of within-person variables

| Variable | $M$ | SD | 1 | 2 | 3 |
|---|---|---|---|---|---|
| 1. Momentary self-esteem | 6.28 | 2.49 | | | |
| 2. Time | 16.50 | 10.06 | 0.03* | | |
| 3. Weekend | 0.29 | 0.45 | 0.03* | 0.00 | |
| 4. Interaction partner present | 0.50 | 0.50 | 0.08* | $-0.14$* | $-0.04$* |

Weekend and interaction partner present were dummy coded (0 = weekday, 1 = weekend; 0 = no interaction, 1 = interaction partner present).
$M$ mean, $SD$ standard deviation.
*indicates $p < 0.05$.

## Table 2 | Means, standard deviations, and intercorrelations of between-person variables

| Variable | $M$ | SD | 1 | 2 | 3 | 4 | 5 | 6 | 7 | 8 |
|---|---|---|---|---|---|---|---|---|---|---|
| 1. MSE BP | 6.14 | 1.90 | | | | | | | | |
| 2. Trait self-esteem | 4.58 | 1.56 | 0.70* | | | | | | | |
| 3. Reappraisal | 5.49 | 2.35 | 0.35* | 0.43* | | | | | | |
| 4. ES | 5.03 | 2.71 | $-0.35$* | $-0.47$* | $-0.26$* | | | | | |
| 5. Reflection | 6.33 | 2.34 | 0.19* | 0.14* | 0.38* | $-0.27$* | | | | |
| 6. Social sharing | 6.34 | 2.81 | 0.18* | 0.23* | 0.13* | $-0.53$* | 0.34* | | | |
| 7. Age | 16.83 | 1.41 | 0.09 | 0.15* | 0.12* | $-0.16$* | 0.11* | 0.13* | | |
| 8. Gender | 0.18 | 0.39 | 0.19* | 0.24* | 0.13* | 0.01 | $-0.01$ | $-0.16$* | 0.14* | |
| 9. Sample | 0.48 | 0.50 | $-0.19$* | $-0.20$* | $-0.15$* | 0.24* | $-0.18$* | $-0.19$* | $-0.63$* | $-0.16$* |

Gender and the sample of the original study were dummy coded (0 = female, 1 = male; 0 = Study 1, 1 = Study 2).
$M$ mean, $SD$ standard deviation, $MSE$ $BP$ momentary self-esteem between-person (for each individual, momentary self-esteem was averaged across within person measurements), $ES$ expressive suppression.
*indicates $p < 0.05$.

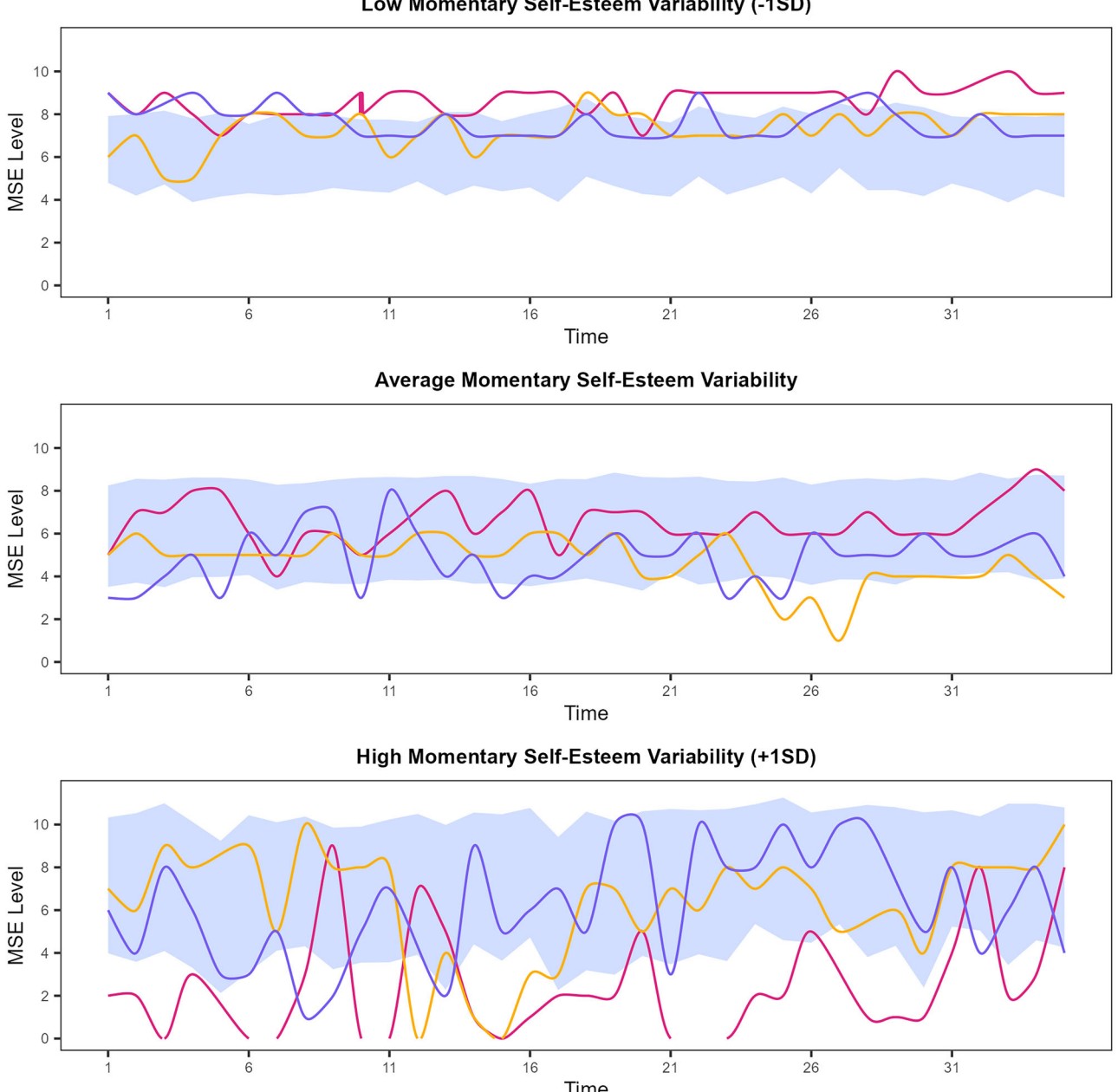

**Fig. 1 | Adolescents' momentary self-esteem level and variability across time.** MSE momentary self-esteem, SD standard deviation. This figure illustrates momentary self-esteem level and variability across time. Based on the degree of variability in their momentary self-esteem, participants were grouped into one of three subgroups: low (1 SD below the mean), average (at the mean), and high (1 SD above the mean) momentary self-esteem variability. In each subfigure, the blue ribbons represent the mean momentary self-esteem ±1 SD, calculated for each of the 35 ESM ratings across participants in the respective subgroups. The individual lines represent the trajectories of three exemplary participants from each subgroup, selected according to the criteria of low number of missing data and typicality for their group.

CI = [−0.09, 0.21]). When using the relative coding (individual means compared to the sample means as indicators of (non)usage), a higher number of used strategies was statistically significantly associated with higher momentary self-esteem levels ($\gamma = 0.28$; 95% CI = [0.03, 0.52]), but not with self-esteem variability ($\omega = 0.03$; 95% CI = [−0.11, 0.16]). Thus, if adolescents used certain strategies similarly often or even more than their peers, they reported higher momentary self-esteem levels. The effect did not remain robust when controlling for covariates ($\gamma = 0.22$; 95% CI = [−0.02, 0.46]).

In our second exploratory analysis, we extended the MELSMs of the main analyses by including adolescents' levels of trait self-esteem as a between-person variable predicting momentary self-esteem level and variability (see Supplementary Table 6 in the OSM). These models indicated that adolescents with higher trait self-esteem reported higher momentary self-esteem levels ($\gamma = 0.85$; 95% CI = [0.75, 0.94]; from the model including reappraisal) and experienced less variability ($\omega = -0.16$; 95% CI = [−0.23, −0.10]; from the model including reappraisal) in momentary self-esteem across daily situations. Regarding ER strategies, reflection remained a significant positive predictor of higher momentary self-esteem levels ($\gamma = 0.07$; 95% CI = [0.02, 0.13]), while social sharing emerged as a positive predictor for momentary self-esteem variability ($\omega = 0.04$; 95% CI = [0.00, 0.07]) when controlling for trait self-esteem.

**Table 3 | Momentary self-esteem regressed on individual ER strategies**

| | Reappraisal | | Expressive suppression | | Reflection | | Social sharing | |
|---|---|---|---|---|---|---|---|---|
| | Est. | 95% CI | Est. | 95% CI | Est. | 95% CI | Est. | 95% CI |
| MSE level intercept | **6.17** | [5.99, 6.35] | **6.17** | [5.99, 6.35] | **6.17** | [5.98, 6.36] | **6.17** | [5.98, 6.36] |
| MSE variability intercept | **0.79** | [0.69, 0.89] | **0.79** | [0.69, 0.89] | **0.79** | [0.69, 0.89] | **0.79** | [0.69, 0.89] |
| Linear time trend | 0.00 | [0.00, 0.01] | 0.00 | [0.00, 0.01] | 0.00 | [0.00, 0.01] | 0.00 | [0.00, 0.01] |
| MSE level | | | | | | | | |
| ER strategy | **0.28** | [0.21, 0.36] | **−0.25** | [-0.31, -0.18] | **0.16** | [0.08, 0.24] | **0.12** | [0.05, 0.18] |
| MSE variability | | | | | | | | |
| ER strategy | **−0.05** | [−0.09, −0.01] | **0.04** | [0.01, 0.08] | −0.03 | [−0.07, 0.02] | 0.01 | [−0.02, 0.05] |
| Residual variances | | | | | | | | |
| MSE level | **3.10** | [2.68, 3.60] | **3.09** | [2.67, 3.59] | **3.38** | [2.93, 3.93] | **3.41** | [2.95, 3.96] |
| MSE variability | **0.87** | [0.73, 1.02] | **0.87** | [0.73, 1.02] | **0.88** | [0.74, 1.03] | **0.88** | [0.75, 1.04] |
| Time | **0.00** | [0.00, 0.00] | **0.00** | [0.00, 0.00] | **0.00** | [0.00, 0.00] | **0.00** | [0.00, 0.00] |
| Explained variance | | | | | | | | |
| $\Delta R^2_{level}$ | 0.12 | | 0.12 | | 0.04 | | 0.03 | |
| $\Delta R^2_{variability}$ | 0.01 | | 0.01 | | 0.00 | | 0.00 | |

Estimates reflect unstandardized model results. Results are based on $n = 8349$ observations nested in 408 individuals. Estimates concerning momentary self-esteem variability are on a logarithmic scale. Estimates in bold font have 95% credible intervals not including zero and are considered statistically significant. $\Delta R^2_{Level}$ and $\Delta R^2_{Variability}$ represent the proportion of variance that can be explained by all model predictors relative to a null model.
*MSE* momentary self-esteem.

## Discussion

The present study investigated the role of using four central ER strategies—reappraisal, expressive suppression, reflection, and social sharing—in adolescents' momentary self-esteem level and variability in daily life. Our analyses, based on a large sample of adolescents, revealed two key findings: first, adolescents who regulated their emotions to a greater extent using reappraisal, reflection, and social sharing as ER strategies reported higher momentary self-esteem levels. In contrast, adolescents who tended to engage more frequently in expressive suppression reported lower momentary self-esteem levels in their daily lives. Second, expressive suppression emerged as robust predictor of self-esteem variability with adolescents using expressive suppression more often illustrating higher self-esteem variability. Overall, the effects were comparatively larger for momentary self-esteem levels, as reflected in the higher proportions of explained variance compared to variability. In the following, we discuss four key implications of these results that stood out.

First, among the ER strategies, reappraisal and expressive suppression were the strongest predictors of adolescents' momentary self-esteem levels. Thus, the tendency to use reappraisal as an ER strategy, which entails that people think of emotional situations as learning environments, appears to be linked to more positive self-evaluations across daily situations. In contrast, the tendency to regulate emotions through expressive suppression, such as habitually covering up emotions, seems to be associated with more negative self-evaluations. This pattern aligns with our hypotheses and reflects existing literature on the relation between these ER strategies and trait self-esteem in adults[33]. Our robustness analyses further emphasized that although effects of all ER strategies persisted when controlling for covariates, only effects of reappraisal and expressive suppression remained robust, when considering all ER strategies simultaneously. What could explain such inconsistency? On the one hand, the positive zero-order correlations between reflection and social sharing with reappraisal and the negative with expressive suppression could suggest that adolescents who regulate their emotions through reflection and social sharing are generally more likely to use adaptive and less likely to use maladaptive ER strategies. At the same time, the conjoint consideration of the ER strategies suggests that effects on momentary self-esteem levels may be primarily driven by reappraisal and expressive suppression. On the other hand, while we generally found positive effects of social sharing on momentary self-esteem, previous results have been more mixed. For example, excessive problem discussions and

dwelling on negative feelings with friends can emerge as a downside of social sharing and is associated with lower self-esteem[57,58].

Second, despite the detrimental role of self-esteem variability on mental health[7–10], our results indicate that predicting momentary self-esteem variability is challenging. As the one robust predictor, expressive suppression was consistently associated with momentary self-esteem variability. Adolescents who tended to cover up their emotions were reporting more fluctuations in the way they evaluated themselves. Thus, this strategy is not only a consistent negative predictor of self-esteem levels but seems to be associated with the detrimental pattern of more variability. Our findings provide an additional insight into variability by revealing an association between higher momentary self-esteem levels and lower variability among adolescents. A more positive self-view is linked to a greater consistency in how adolescents perceive their self-worth. This aligns with adult findings suggesting that lower self-esteem levels are linked to more variability, which has been related to a "fragile" self[59]. These findings further emphasize that we need to better understand the underlying mechanisms of within-person variability in momentary self-esteem. Thus, future research should investigate the factors and mechanisms contributing to these fluctuations, especially in the critical developmental phase of adolescence.

Third, our exploratory analyses suggested that a larger number of ER strategies used by adolescents might be linked to higher levels of momentary self-esteem. It should be noted, however, that the operationalization of the number of ER strategies used in this study was somewhat limited in that whether an ER strategy was considered as used was artificially derived into a composite score. Yet, the finding aligns well with existing literature suggesting that a greater repertoire of ER strategies is adaptive in adolescence[32].

Fourth, in a final exploratory step, we examined whether ER strategies predicted momentary self-esteem levels and variability over and above trait self-esteem. Given the substantial overlap between trait and state self-esteem[3], we consider this step to be a conservative test. Yet, even in these analyses, we found substantial effects on both self-esteem levels and variability. Most notably, in these additional models, using social sharing was linked to higher momentary self-esteem variability. What might explain this link between social sharing and momentary self-esteem variability? On the one hand, the saying "A trouble shared is a trouble halved" would suggest that social sharing has favorable effects, which aligns with the level effects observed in our simple models. On the other hand, social sharing might involve critical feedback or the rise of new emotions through joint reflection,

potentially leading to self-related uncertainty and, in turn, greater variability in momentary self-esteem. This perspective fits with sociometer theory that views self-esteem as a monitor of social feedback[38]. As such, we want to emphasize that more research is needed to understand what level of variability should be considered (mal)adaptive and whether there is something like a "sweet spot"[3]. Similar ideas have been discussed in research on emotion dynamics, as both excessive variability and emotional inertia might be linked to maladaptive outcomes[60]. When it comes to effects of trait self-esteem, the established overlap between trait and state self-esteem explains the amount of explained variance in momentary levels, yet even trait self-esteem explains only limited amounts of variance in variability. This further emphasizes the open question: what predicts interindividual differences in momentary self-esteem variability?

Taken together, our results suggest that ER is relevant for momentary self-esteem level and variability in adolescents' daily lives. Furthermore, the differential associations between ER strategies and self-esteem dynamics highlight the value of fostering adaptive ER from a young age onward. Through the lens of sociometer theory[38], our results indicate that ER plays a key role in self-esteem through social belonging, at least during adolescence. Notably, previous research has shown that adolescent ER varies across social contexts—for example, adolescents are less effective at using reappraisal in social contexts, particularly when they are sensitive to rejection[61], and they are more likely to suppress their emotional expressions around peers than around family members[62]. Thus, in the context of the broad literature, our findings highlight that learning to regulate one's emotions in social situations represents a key developmental task that is well worth mastering.

### Limitations

The current study has several strengths, including the insights into the daily lives of adolescents using a large sample and the application of modern statistical methods that provide insights into adolescents' momentary self-esteem. However, we also note several limitations. First, the correlational design of the study does not allow for any causal conclusions to be drawn and might be affected by potential confounding variables that were not included in our set of covariates. For example, we reported that the use of social sharing contributes to higher levels of momentary self-esteem. However, it may also be the case that adolescents with higher momentary self-esteem levels are more likely to be in relationships that provide a supportive environment for sharing their emotions, which in turn may facilitate a stronger tendency to use social sharing as ER strategy. Second, our sample consisted mostly of female participants. As prior research often reports gender differences in ER[63], and self-esteem[64], our results might not generalize to other adolescent populations. Future research should include more diverse samples of adolescents in terms of gender and other socio-demographic variables (e.g., socioeconomic status). Third, we identified some differences between the participants in our original studies, which were accounted for in the statistical models. Fourth, participants received ESM prompts at fixed times. While fixed sampling schemes may help participants integrate prompts into their school-day and daily routines, it can reduce ecological validity if they anticipate beeps and adapt their behavior accordingly (e.g., seeking out quiet environments to answer the questionnaires[65]).

### Outlook

Moving forward, future research could explore the link between ER strategies and momentary self-esteem components in more depth. While our results suggest an association between ER and between-person differences in self-esteem variability, our analyses do not provide insights into the underlying mechanisms that explain why some adolescents vary more in their self-esteem than others. It is possible that adolescents' ER is linked to how randomly their self-esteem fluctuates. Alternatively, ER may be involved in their self-esteem reactivity to specific situations[3] or their exposure to situations that trigger frequent self-esteem changes. Therefore, incorporating situational predictors represents the essential next step to disentangle the contributions of individual and situational factors to self-esteem variability. In addition, future research could investigate how self-esteem dynamics relate to adolescents' effectiveness of ER (i.e., ER ability[66]), their ability to apply ER strategies according to contextual demands and personal goals (i.e., ER flexibility[67]), and their capacity to use multiple ER strategies in response to a single emotion-eliciting situation (i.e., emotion polyregulation[68]). Further, the observed association between momentary self-esteem levels and variability highlights the need to better understand the processes linking these two components, which in turn may provide insight into predictors of variability.

## Conclusions

Our study showed that ER strategies are associated with interindividual differences in momentary self-esteem levels and momentary self-esteem variability in adolescence. Reappraisal and expressive suppression strategies were the most robust ER strategies predicting momentary self-esteem levels, emphasizing the importance of helping adolescents to show and learn from their new emotional experiences in this dynamic period. Additionally, adolescents with a higher momentary self-esteem level reported lower momentary self-esteem variability, emphasizing the importance of supporting adolescents to develop a positive view of themselves. Future research needs to further invest in understanding sources and mechanisms fueling momentary self-esteem levels and variability in daily life to help educators, clinicians, families, and adolescents themselves to navigate this critical developmental period.

## Data availability

All data used to generate results of the present study are available on OSF (https://osf.io/9ntck/).

## Code availability

Data cleaning, preparation and descriptive statistics were conducted in R version 4.4.2 (R Core Team) using R Studio version 2024.12.1 + 563 (Posit Team). Statistical models were estimated in Mplus version 8.5[69]. The code of the presented analyses is publicly available on OSF (https://osf.io/9ntck/).

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

## Acknowledgements
This research was funded by the German Research Foundation (DFG) via Grant WA 3509/3-1 awarded to Jenny Wagner. The funders had no role in study design, data collection and analysis, decision to publish or preparation of the manuscript. We further acknowledge financial support from the Open Access Publication Fund of the University of Hamburg.

## Author contributions

Dennis Warnholtz: Conceptualization, Methodology, Formal Analysis, Visualization, Writing—Original Draft, Writing-Reviewing and Editing. Larissa Lucia Wieczorek: Methodology, Writing—Reviewing and Editing. Eva Bleckmann: Writing—Reviewing and Editing. Jenny Wagner: Conceptualization, Funding acquisition, Writing—Reviewing and Editing, Supervision.

## Funding

## Competing interests
The authors declare no competing interests.
