## [Transparent Peer Review file · Communications Psychology]

Trait Emotion Regulation Predicts Momentary Self-Esteem Level and Variability in Adolescents' Daily Lives

Corresponding Author: Mr Dennis Warnholtz

Version 0:

Decision Letter:

Dear Mr Warnholtz,

Thank you for your patience during the peer-review process. Your manuscript titled "Dynamics in Everyday Life: Linking Adolescents' Emotion Regulation to Momentary Self-Esteem Levels and Variability" has now been seen by 2 reviewers, and I include their comments at the end of this message. They find your work of interest but raised some important points. We are interested in the possibility of publishing your study in Communications Psychology, but would like to consider your responses to these concerns and assess a revised manuscript before we make a final decision on publication.

We therefore invite you to revise and resubmit your manuscript, along with a point-by-point response to the reviewers. Please highlight all changes in the manuscript text file.

Editorially, we consider it important that the revised manuscript addresses the reviewers' methodological and analytic concerns and, in particular, provides support for the validity of the ESM self-esteem measure. Please remove all causal language and provide information regarding informed consent.

I am attaching an Editorial Requests Table that details critical reporting requirements for the revised manuscript. Please attend to each item and ensure your manuscript is fully compliant. If your revised manuscript is not aligned with these requests on major issues, such as those concerning statistics, it may be returned to you for further revisions without re-review.

Please submit the following items:

- Revised manuscript
- Point-by-point response to the referees' comments
- Cover letter (as a separate document)
- <https://www.nature.com/documents/nr-reporting-summary.zip>>Nature Research Reporting Summary
- <https://www.nature.com/documents/nr-editorial-policy-checklist.pdf>>Editorial Policy Checklist
- Completed Editorial Request Table (attached).

via this link: Link Redacted .

Additional guidance is available in our style and formatting guide Communications Psychology formatting guide.

Best regards,

Jennifer Bellingtier

Jennifer Bellingtier, PhD
Senior Editor
Communications Psychology

REVIEWER EXPERTISE:

Reviewer #1 Emotion Regulation, Intensive longitudinal designs
Reviewer #2 Emotion Regulation, Self-esteem, Adolescent development

REVIEWER REPORTS:

Reviewer #1 (Remarks to the Author):

I appreciated the opportunity to review this manuscript and enjoyed reading it. I believe that the manuscript makes a meaningful contribution to the field of adolescent self-esteem dynamics in daily life. Overall, the manuscript is well-written, concise, and clear. Diligence was done on methodological and data analytic decisions. The conducted statistical analyses are well-matched to the research objectives and findings were clearly communicated. Finally, a strong commitment was done to open practices and transparency which is commendable. I went carefully through the manuscript, the preregistration, and the materials put online. I have only a few comments and suggestions that mainly concern clarification.

1) Introduction section. The introduction establishes the relevance of self-esteem variability in relation to psychopathology. However, I suggest briefly acknowledging theoretical perspectives that highlight potential curvilinear associations of variability with outcomes, such as the possibility that both very low and very high variability may be maladaptive. Prior work in affect variability (e.g., Houben et al., 2015, <http://dx.doi.org/10.1037/a0038822>) supports such interpretations. The authors wrote something similar in the Discussion section, but perhaps a (briefly) mention of this in the introduction could provide more context for the reader.

2) Introduction section. While reading this section I had the assumption that ER would be assessed at the momentary level until I read the Methods section. Could the authors try to clarify that is the trait or tendency that is included? This could be done when discussing the research objectives or sooner.

3) Method section. Concerning the item used to assess momentary self-esteem, was this item based on previous work (in ESM) and has it been tested or validated with adolescents? I wonder since the item itself seems to be not that easy as it requires self-reflection and emotional awareness, which is in ongoing development during adolescence and may be challenging to report.

4) Method section. The authors mention that participants received ESM prompts during fixed times. The use of fixed time schedules can be convenient for participants (i.e., to avoid sending surveys during inconvenient times), but it can also lower ecological validity as you can anticipate the next prompt. Can this decision be clarified in the manuscript or is it mentioned in the accompanying codebooks?

5) Data analyses section. Concerning the computed indicator of the number of ER strategies, the authors explained the use

of absolute and relative coding. The authors also acknowledge the limitations of this indicator. However, it would be helpful to briefly explain the rationale behind the chosen thresholds (e.g., “score below 3” or “1 SD below the mean”). Are these decisions based on prior work or “arbitrary” due to a lack of established cutoffs?

6) Results section. A detail but the results are written as: “Thus, adolescents who used reappraisal more often exhibited lower self-esteem variability, whereas those who employed expressive suppression more often experienced greater fluctuations in their self-esteem”. These are statements are about frequency in use but given the way emotion regulation was worded and rated, it could be more accurate or more clear to talk about tendency: “adolescents who indicated a stronger tendency ...”.

7) In addition to the above comments, I had some minor corrections.

a. There are a few minor mistakes in the output tables but these do not in any way alter any conclusions. For example, in Table 3, concerning the relation of momentary self-esteem predicted by reflection, the confidence interval should be [0.08, 0.24] instead of [0.08, 0.23]. Another example, On the OSF Table A5, some estimates are put in bold but contain 0 in the CI such as estimate 0.48 with CI [-0.02, 0.98]. Again, these mistakes are very minor but I would recommend checking the output.

b. I noticed some occurrences of causal language. For example, in the Discussion section: “emotional situations as learning environments, can apparently boost their self-evaluations”. I also noticed that the term “relationship” (connections between (non)-human animals) should sometimes be “relation” (connection between variables). For example: “This pattern aligns with our hypotheses and reflects existing literature on the relationship” -> relation

Reviewer #2 (Remarks to the Author):

Dear authors, thank you for giving me the opportunity to review your manuscript which I really liked. I think your research is likewise interesting as it is important and your data feels completely adequate, your analyses profound. View my remarks with this general evaluation in mind. I hope my remarks are clear and constructive that they might help to make the manuscript even better.

Dear authors, thank you for giving me the opportunity to review your manuscript which I really liked. I think your research is likewise interesting as it is important and your data feels completely adequate, your analyses profound. View my remarks with this general evaluation in mind. I hope my remarks are clear and constructive that they might help to make the manuscript even better.

Introduction:

- I like the introduction. Precise and on point, explaining research gaps and research interest
- In my opinion the introduction lacks a theoretical foundation: At the moment, the research questions are widely based on this paragraph:
Since momentary self-esteem is fundamentally an emotional evaluation—seeing oneself positively or negatively—a crucial factor in understanding self-esteem levels and variability in everyday life is how well adolescents can manage their emotions.
- Is there theory guiding your research questions? Otherwise – if exploratory, I would suggest to make a stronger argument on your hypothesized relations.
- To my liking, your presentation of the empirical evidence feels a bit unprecise:
 - o You state that re-appraisal linked to mental-health, well-being
 - o Reflection is not elaborated on; yet, you switch over to the other „two remaining“
 - o Social sharing – only empirical evidence with regards to emotional recovery
 - o Then you switch over to developmental aspects of ER
 - o At the moment, your only presented evidence on the relationship with ER and self-esteem comes from this sentence: Research suggests that the use of more adaptive strategies and fewer maladaptive strategies is linked to higher trait self-esteem in adults (Gross & John, 2003).
- I would recommend
 - o Start defining the four strategies
 - o Highlight developmental aspects of the four strategies, if available – if not, state that (to address the specificity of your sample)
 - o Empirical evidence on the relationship between the ER strategies and self-esteem; if not, state that and continue with empirical evidence of related-constructs
 - o In your introduction you highlight the differentiation between level and variability, which I really like and agree with your argument: However, this distinction is nowhere else to be found: Make your research contribution more evident when you highlight that existing studies solely focus on level differences and ignore intra-individual differences
- I think the research goals paragraph at the end is too vague and superficial. Please elaborate on your goals and hypotheses and state hypotheses if theory/ empirical evidence allows it or make it transparent that you address some research questions exploratorily.

Method

- Measures

- o Can you justify your ESM measure from the literature? Is it a validated measure? Where have you taken it from?
- o What is the rsb coefficient? I don't know that one
- o Perhaps that's me and I know many other people ignore that too, but from a measurement POV, you report McD's Omega

highlighting a congeneric measurement model, yet you simply average items.

- Statistical Analyses

o Can you justify your choice of model? I am not familiar with MELMS. Perhaps keep that in mind, when reading my remark: With your data my guess was that you would conduct a DSEM model which incorporate autoregressive components. At the moment you have included robustness checks which are fine, but I guess the autoregressive component of momentary self-esteem might be quite influential. This shouldn't be a critique and I know that DSEM makes it harder to address the variability component of your analyses. I simply got stuck on the question: Would your effects be significant if your model incorporated the autoregressive effects of MSEt-1 on MSEt

Results

o Perhaps state that you are referring to ICC(1) here which examines the allocation of variance across different levels whereas the ICC(2) you mentioned in the measures section is a measure of reliability

- Discussion I really like your discussion in general and my remarks are only an re-iteration of the points I have made concerning your introduction

o I think, a clear reiteration of research questions and hypotheses would be nice.

o You do a great job in discussion and evaluating your effects, but I think the discussions falls short of a paragraph on the theoretical impact of the research.

I checked output files within the OSF repository. And there are probably multiple copy&paste mistakes in Table 3 of the manuscript.

For example:

File 1.2 melms_mse_suppression.out

R-squared: 0.068 in the file; 0.12 in Table:

Variability intercept: 0.846 in the file; 0.79 in the Table

File 1.2 melms_mse_reappraisal.out

R-squared: 0.066 in the file; 0.12 in Table:

I haven't checked all files; but the authors should check all again and highlight differences and whether these differences impact conclusions/discussions/.

* TRANSPARENT PEER REVIEW: Communications Psychology uses a transparent peer review system. This means that we publish the editorial decision letters including Reviewers' comments to the authors and the author rebuttal letters online as a supplementary peer review file. However, on author request, confidential information and data can be removed from the published reviewer reports and rebuttal letters prior to publication. If your manuscript has been previously reviewed at another journal, those Reviewers' comments would not form part of the published peer review file.

If you experience problems in linking your ORCID, please contact the Platform Support Helpdesk.

Version 1:

Decision Letter:

Dear Mr Warnholtz,

Thank you for your patience during the peer-review process. Your manuscript titled "Trait Emotion Regulation Predicts Momentary Self-Esteem Level and Variability in Adolescents' Daily Lives" has now been seen by 2 reviewers, and I include their comments at the end of this message. They find your revisions have improved the manuscript, but a few concerns remain. We are interested in the possibility of publishing your study in Communications Psychology, but would like to consider your responses to these concerns and assess a revised manuscript before we make a final decision on publication.

We therefore invite you to revise and resubmit your manuscript, along with a point-by-point response to the reviewers. Please highlight all changes in the manuscript text file.

Editorially, we consider it important that the revised manuscript provide additional evidence that the findings are robust to the inclusion of within-person predictors or acknowledge this as a limitation in the manuscript. Furthermore, it is imperative that all deviations from the preregistration be disclosed in the main text, and all preregistered analyses reported.

I am attaching an Editorial Requests Table that details critical reporting requirements for the revised manuscript. Please attend to each item and ensure your manuscript is fully compliant. If your revised manuscript is not aligned with these requests on major issues, such as those concerning statistics, it may be returned to you for further revisions without re-review.

Please submit the following items:

- Revised manuscript
- Point-by-point response to the referees' comments
- Cover letter (as a separate document)
- <https://www.nature.com/documents/nr-reporting-summary.pdf>>Nature Research Reporting Summary
- Completed Editorial Request Table (attached).

via this link: Link Redacted .

Additional guidance is available in our style and formatting guide <https://www.nature.com/documents/commpsychol-style-formatting-guide-accept.pdf>>Communications Psychology formatting guide.

Best regards,

Jennifer Bellingtier

Jennifer Bellingtier, PhD
Senior Editor
Communications Psychology

REVIEWER EXPERTISE:

Reviewer #1 Emotion Regulation, Intensive longitudinal designs
Reviewer #2 Emotion Regulation, Self-esteem, Adolescent development

REVIEWER REPORTS:

Reviewer #1 (Remarks to the Author):

I appreciate the opportunity to review the revised manuscript. I went to the manuscript and I have carefully reviewed the changes. Overall, I deem the paper to be improved, particularly in clarity. The authors have adequately addressed all of my comments. Personally I would recommend the manuscript for acceptance.

Reviewer #2 (Remarks to the Author):

Thank you very much for your thorough revision. I am basically happy for most parts of the revision - however, there is one point where your rebuttal has not convinced me and I think that this is still an issue.

Comment 8 —

"As our models include only between-person predictors, accounting for autoregressive components should not alter the estimated associations between ER strategies and interindividual differences in momentary self-esteem levels or variability."
"

I think this is simply not true or it ignores the point that I was trying to make. I try better. Your between-level predictors - the scale part is highly dependent on the within part, so this is not a pure between-level issue. Specifically, I am referring to the random residual variance. Momentary self-esteem fluctuates - and some adolescents fluctuate more strongly around their mean than others. In the within part, you partial out the trajectory proportion within your data, but the time predictor is insignificant on average. So there is not much to partial out. Within your model you treat these small and strong fluctuations of adolescents as kind of trait variable - meaning that it is assumed that we have an Adolescent A who, across days, fluctuates very strongly in his/her MSE around their personal mean while Adolescent B who is very stable in his/her MSE. So Person A is the unstable MSE person, while person B is the stable one.

The model you propose basically ignores the fact that there might be reasons that are inherent to the situation where MSE was captured that make their MSE fluctuate more/ less strongly. Predictors on the within-level will reduce the residual variance, so your between level correlations will be affected by within-part. Don't get me wrong - I see your point and agree with the fact that there are adolescents in your sample that will be more or less stable in their MSE regardless of the situation, but currently situation specific predictors are ignored which leaves much more variance proportions to potentially correlate with other variables on the between level. The autoregressive example is just one example among others.

Additionally, in your discussion you write about robust predictors of variability. In a statistical sense, I of course agree with that conclusion, but from a practical point of view we are talking about borderline-significance. Both significant effects nearly touch the zero in the CIs. Paired with the small regression coefficient and the small explained variance, I think these results should be expressed more reasonably in terms of their significance.

* TRANSPARENT PEER REVIEW: Communications Psychology uses a transparent peer review system. This means that we publish the editorial decision letters including Reviewers' comments to the authors and the author rebuttal letters online as a supplementary peer review file. However, on author request, confidential information and data can be removed from the published reviewer reports and rebuttal letters prior to publication. If your manuscript has been previously reviewed at another journal, those Reviewers' comments would not form part of the published peer review file.

If you experience problems in linking your ORCID, please contact the Platform Support Helpdesk.

Version 2:

Decision Letter:

Dear Mr Warnholtz,

Your manuscript titled "Trait Emotion Regulation Predicts Momentary Self-Esteem Level and Variability in Adolescents' Daily Lives" has now been editorially reviewed, and I am delighted to say that we are happy, in principle, to publish a suitably revised version in Communications Psychology.

We therefore invite you to revise your paper one last time to address the remaining concerns of our reviewers and a list of editorial requests. At the same time we ask that you edit your manuscript to comply with our format requirements and to maximise the accessibility and therefore the impact of your work.

EDITORIAL REQUESTS:

SUBMISSION INFORMATION:

OPEN ACCESS:

*** TRANSPARENT PEER REVIEW:** Communications Psychology uses a transparent peer review system. On author request, confidential information and data can be removed from the published reviewer reports and rebuttal letters prior to publication. If you are concerned about the release of confidential data, please let us know specifically what information you would like to have removed. Please note that we cannot incorporate redactions for any other reasons.

*** CODE AVAILABILITY:** All Communications Psychology manuscripts must include a section titled "Code Availability" at the end of the methods section. We require that the custom analysis code supporting your conclusions is made available in a publicly accessible repository at this stage; please choose a repository that generates a digital object identifier (DOI) for the code; the link to the repository and the DOI must be included in the Code Availability statement. Publication as Supplementary Information will not suffice.

*** DATA AVAILABILITY:**

Link Redacted

Best regards,

Jennifer Bellingtier

Jennifer Bellingtier, PhD
Senior Editor
Communications Psychology

Reviewer Reports

Reviewer 1

R1 #1: *I appreciated the opportunity to review this manuscript and enjoyed reading it. I believe that the manuscript makes a meaningful contribution to the field of adolescent self-esteem dynamics in daily life. Overall, the manuscript is well-written, concise, and clear. Diligence was done on methodological and data analytic decisions. The conducted statistical analyses are well-matched to the research objectives and findings were clearly communicated. Finally, a strong commitment was done to open practices and transparency which is commendable. I went carefully through the manuscript, the preregistration, and the materials put online. I have only a few comments and suggestions that mainly concern clarification.*

Response: Thank you for this positive evaluation of our manuscript and for your constructive feedback.

R1 #2: *Introduction section. The introduction establishes the relevance of self-esteem variability in relation to psychopathology. However, I suggest briefly acknowledging theoretical perspectives that highlight potential curvilinear associations of variability with outcomes, such as the possibility that both very low and very high variability may be maladaptive. Prior work in affect variability (e.g., Houben et al., 2015, <http://dx.doi.org/10.1037/a0038822>) supports such interpretations. The authors wrote something similar in the Discussion section, but perhaps a (briefly) mention of this in the introduction could provide more context for the reader.*

Response: Thank you for your suggestion on the theoretical background. We agree that the potential curvilinear associations of variability in self-esteem are important. However, since the focus of our research is on predictors of variability rather than its outcomes, we feel more confident highlighting this point in the discussion (page 21):

“As such, we want to emphasize that more research is needed to understand what level of variability should be considered (mal)adaptive and whether there is something like a “sweet spot” (Wagner et al., 2024). **Similar ideas have been discussed in research on emotion dynamics, as both excessive variability and emotional inertia might be linked to maladaptive outcomes (Houben et al., 2015).**”

R1 #3: *Introduction section. While reading this section I had the assumption that ER would be assessed at the momentary level until I read the Methods section. Could the authors try to clarify that is the trait or tendency that is included? This could be done when discussing the research objectives or sooner.*

Response: Thank you for this important remark. We agree that it was not entirely clear at which level we assessed ER in the current study and revised several parts of the manuscript to clarify that ER strategy use was assessed as a trait-like tendency. First, our new manuscript title, “Trait Emotion Regulation Predicts Momentary Self-Esteem Level and Variability in Adolescents’ Daily Lives”, more clearly reflects the trait-level focus of our assessment of ER. Further, in the abstract we now state:

“The current study examines **how the trait-level use of four emotion regulation strategies**—reappraisal, reflection, expressive suppression, social sharing—**is associated with the** level and variability of momentary self-esteem in the developmentally critical period of adolescence.”

Similarly, the revised introduction of our research objectives now reads (page 6):

“The present research investigates how trait ER strategy use relates to momentary self-esteem level and variability in adolescence.”

R1 #4: *Method section. Concerning the item used to assess momentary self-esteem, was this item based on previous work (in ESM) and has it been tested or validated with adolescents? I wonder since the item itself seems to be not that easy as it requires self-reflection and emotional awareness, which is in ongoing development during adolescence and may be challenging to report.*

Response: Thank you for your comment regarding the origin and validity of the item used to assess momentary self-esteem. The item was adapted from the Rosenberg Self-Esteem Scale (Rosenberg, 1965), which was originally developed for use with adolescent populations. The item in question has successfully been used in adolescent samples (e.g.; Bleckmann et al., 2023). Moreover, the ICC(2) indicated good reliability, further suggesting that the item was suitable to measure momentary self-esteem in our sample. We have now added the appropriate reference and explanation in the revised Methods section on page 8.

“Momentary self-esteem was assessed with a single item (“All things considered, how content are you with yourself right now?”), which was adapted from the Rosenberg Self-Esteem Scale (Rosenberg, 1965).”

R1 #5: *Method section. The authors mention that participants received ESM prompts during fixed times. The use of fixed time schedules can be convenient for participants (i.e., to avoid sending surveys during inconvenient times), but it can also lower ecological validity as you can anticipate the next prompt. Can this decision be clarified in the manuscript or is it mentioned in the accompanying codebooks*

Response: As you correctly noted, the two original studies featured in our research used a fixed sampling scheme. Fixed intervals are useful for assessments in adolescent populations, as they allow participants to plan for completing the questionnaires and thus help integrate them better into their daily routines (Vachon et al., 2019; van Roekel et al., 2019). While ESM designs with fixed timings are a common practice (e.g., Wrzus & Neubauer, 2023), we agree that they may reduce ecological validity in some cases and added a note on this in the limitation section on page 22:

“Fourth, participants received ESM prompts at fixed times. While fixed sampling schemes may help participants integrate prompts into their school-day and daily routines, it can reduce ecological validity if they anticipate beeps and adapt their behavior accordingly (e.g., seeking out quiet environments to answer the questionnaires; Vachon et al., 2019).”

R1 #6: *Data analyses section. Concerning the computed indicator of the number of ER strategies, the authors explained the use of absolute and relative coding. The authors also acknowledge the limitations of this indicator. However, it would be helpful to briefly explain the rationale behind the chosen thresholds (e.g., “score below 3” or “1 SD below the mean”). Are these decisions based on prior work or “arbitrary” due to a lack of established cutoffs?*

Response: We agree that a justification of our chosen thresholds would improve the comprehensibility and transparency of the manuscript. While our threshold for the absolute coding approach was chosen arbitrarily due to a lack of established cutoffs, the threshold for relative coding was based on a work of Lougheed and Hollenstein (2012). We therefore added a footnote on page 11 that reads:

“The threshold for absolute coding was chosen arbitrarily due to a lack of established cutoffs. The threshold for relative coding was based on Lougheed and Hollenstein (2012).”

R1 #7: *Results section. A detail but the results are written as: “Thus, adolescents who used reappraisal more often exhibited lower self-esteem variability, whereas those who employed expressive suppression more often experienced greater fluctuations in their self-esteem”. These are statements are about frequency in use but given the way emotion regulation was worded and rated, it could be more accurate or more clear to talk about tendency: “adolescents who indicated a stronger tendency ...”.*

Response: Thank you for this helpful remark. To improve clarity, we have revised all sections of the manuscript to make explicit that ER strategy use was assessed as a trait-like tendency. For example, our result section on page 15 now states:

“Adolescents who **indicated a stronger tendency to regulate their emotions through** reappraisal, reflection, and social sharing reported higher levels of momentary self-esteem, whereas expressive suppression was associated with lower levels. In addition, reappraisal was associated with lower self-esteem variability and expressive suppression with higher momentary self-esteem variability. Thus, adolescents **with a stronger tendency to use** reappraisal exhibited lower self-esteem variability, whereas those **who tended** to engage in expressive suppression experienced greater fluctuations in their self-esteem. **Reflection and social sharing were** unrelated to self-esteem variability.”

R1 #8: *There are a few minor mistakes in the output tables but these do not in any way alter any conclusions. For example, in Table 3, concerning the relation of momentary self-esteem predicted by reflection, the confidence interval should be [0.08, 0.24] instead of [0.08, 0.23]. Another example, On the OSF Table A5, some estimates are put in bold but contain 0 in the CI such as estimate 0.48 with CI [-0.02, 0.98]. Again, these mistakes are very minor but I would recommend checking the output.*

Response: Thank you for pointing out these mistakes. The discrepancy in the confidence intervals resulted from the default rounding method of R (“round to even” or “bankers rounding”), which we initially used to prepare our tables. The bold estimate in OSF Table A5 was indeed a formatting error – thank you for catching it. We have revised the rounding procedure to “round half up” and re-checked all model outputs and tables in the paper and the appendix for inconsistencies. None of the changes alter our conclusions in any way.

R1 #9: *I noticed some occurrences of causal language. For example, in the Discussion section: “emotional situations as learning environments, can apparently boost their self-evaluations”. I also noticed that the term “relationship” (connections between (non)-human animals) should sometimes be “relation” (connection between variables). For example: “This*

pattern aligns with our hypotheses and reflects existing literature on the relationship” -> relation.

Response: Thank you for pointing out these language issues. In line with your comment, we revised the manuscript to delete any occurrences implying causal language. The discussion now reads (page 19):

“Thus, the tendency to use reappraisal as an ER strategy, which entails that people think of emotional situations as learning environments, **appears to be linked to more positive self-evaluations across daily situations**. In contrast, the tendency to regulate emotions through expressive suppression, such as habitually covering up emotions, seems to **be associated with** more negative self-evaluations.”

We also replaced all occurrences of the term “relationships” with “relation” or “association” when we referred to abstract relations between variables. For instance, in the discussion section (page 19):

“This pattern aligns with our hypotheses and reflects existing literature on the **relation** between these ER strategies and trait self-esteem in adults (Gross & John, 2003).”

Reviewer 2

R2 #1: *Dear authors, thank you for giving me the opportunity to review your manuscript which I really liked. I think your research is likewise interesting as it is important and your data feels completely adequate, your analyses profound. View my remarks with this general evaluation in mind. I hope my remarks are clear and constructive that they might help to make the manuscript even better.*

Response: We sincerely appreciate your positive and constructive feedback!

R2 #2: *Introduction. I like the introduction. Precise and on point, explaining research gaps and research interest. In my opinion the introduction lacks a theoretical foundation: At the moment, the research questions are widely based on this paragraph: “Since momentary self-esteem is fundamentally an emotional evaluation—seeing oneself positively or negatively—a crucial factor in understanding self-esteem levels and variability in everyday life is how well adolescents can manage their emotions.” Is there theory guiding your research questions? Otherwise – if exploratory, I would suggest to make a stronger argument on your hypothesized relations. To my liking, your presentation of the empirical evidence feels a bit unprecise:*

- *You state that re-appraisal linked to mental-health, well-being*
- *Reflection is not elaborated on; yet, you switch over to the other „two remaining“*
- *Social sharing – only empirical evidence with regards to emotional recovery*
- *Then you switch over to developmental aspects of ER*
- *At the moment, your only presented evidence on the relationship with ER and self-esteem comes from this sentence: Research suggests that the use of more adaptive strategies and fewer maladaptive strategies is linked to higher trait self-esteem in adults (Gross & John, 2003).*

I would recommend:

- *Start defining the four strategies*

- *Highlight developmental aspects of the four strategies, if available – if not, state that (to address the specificity of your sample)*
- *Empirical evidence on the relationship between the ER strategies and self-esteem; if not, state that and continue with empirical evidence of related-constructs*

Response: Thank you for your thorough suggestions regarding the structure of the introduction. We appreciate your ideas and found them very helpful. Accordingly, we have made substantial changes to reflect them. In the revised manuscript, the introduction from page 4 onwards now reads:

“Based on the theoretical notions of the Process Model of Emotion Regulation (Gross, 1998, 2015), a wide range of ER strategies have been explored. The current project examines four strategies: reappraisal (re-evaluating the meaning or relevance of a potentially emotional situation to oneself; Gross, 2015), reflection (engaging in self-attention through curiosity or a desire for self-knowledge; Brans et al., 2013; Trapnell & Campbell, 1999), expressive suppression (inhibition of one's own emotional expression; Gross, 2015), and social sharing (communicating with others about the circumstances of an emotion-eliciting event and one's own emotional reactions; Rimé, 2009). Reappraisal and reflection are considered cognitive ER strategies that are implemented before or during the emergence of an emotion to shape the final emotional response (Brans et al., 2013; Gross, 1998; Parkinson & Totterdell, 1999; Trapnell & Campbell, 1999). By contrast, expressive suppression and social sharing are considered behavioral ER strategies that are used to manage emotional responses that have already unfolded (Brans et al., 2013; Gross, 1998; Parkinson & Totterdell, 1999).

The emotional, cognitive, and social changes during adolescence are pivotal for the development of ER, fostering the use of more cognitive, sophisticated strategies and greater flexibility in their application (Compas et al., 2017; Fombouchet et al., 2023; Zimmer-Gembeck & Skinner, 2011). Reappraisal emerges as an effective strategy protecting adolescents from negative mental health outcomes (Compas et al., 2017; Fombouchet et al., 2023; Silvers, 2022). Although less is known about the use of reflection, adolescents become more capable of attending to and reflecting on their own emotions (Compas et al., 2017; Zimmer-Gembeck & Skinner, 2011). Expressive suppression is already employed by children (Gross & Cassidy, 2019), and its use as behavioral ER strategy increases through adolescence (De France & Hollenstein, 2019). This increased use may reflect the rising importance of peer relationships (Brown & Larson, 2009), as adolescents are more inclined to suppress their emotional expression around their peers than around their families (Wylie et al., 2023). This likely reflects an attempt at impression management (Salisch, 2001) to avoid negative interpersonal consequences associated with the expressing negative emotions (Bengtsson et al., 2022; Wylie et al., 2023). The frequency and role of social sharing in adolescence remains unclear. Social sharing per se does not effectively reduce negative affect, as it can involve excessive problem discussion and dwelling on negative feelings, which is associated with negative mental health outcomes (Abela & Hankin, 2011). While we have discussed each ER strategy individually, it has generally been argued that it is most adaptive to have a broad repertoire of ER strategies in adolescence (Lougheed & Hollenstein, 2012).

Focusing on the links between ER strategy use and self-esteem, existing research has largely focused on trait self-esteem and adult samples, suggesting that reappraisal relates to higher and expressive suppression to lower trait self-esteem (Gross & John, 2003). In adolescence, reappraisal likely mitigates the negative effect of peer victimization on self-esteem (Spyropoulou & Giovazolias, 2024). Additionally,

self-reflection tends to be associated positively with the self-esteem of adults with high self-esteem and negatively among those with low self-esteem (Brown & Brown, 2011). Finally, support seeking through whining and expressing sadness without revealing the underlying cause—a potential facet of social sharing—is linked to lower self-esteem (Don et al., 2019). Conversely, sharing positive emotions is linked to higher-quality social relationships (Gable & Reis, 2010) and may also contribute to higher self-esteem via sociometer processes (Leary & Baumeister, 2000). Yet, it remains unclear how different ER strategies relate to the level and variability of momentary self-esteem in daily life, especially during adolescence.”

R2 #3: *Introduction. In your introduction you highlight the differentiation between level and variability, which I really like and agree with your argument: However, this distinction is nowhere else to be found: Make your research contribution more evident when you highlight that existing studies solely focus on level differences and ignore intra-individual differences.*

Response: Thank you for the suggestion. We agree that reiterating the distinction between level and variability helps clarify our research contribution, and have accordingly added the following sentence to the introduction on page 5:

“Together, previous research highlights the relevance of ER for individual differences in trait self-esteem of adults. However, it remains unclear how different ER strategies relate to momentary self-esteem fluctuations (i.e., level and variability) in daily life, especially in adolescence.”

R2 #4: *Introduction. I think the research goals paragraph at the end is too vague and superficial. Please elaborate on your goals and hypotheses and state hypotheses if theory/empirical evidence allows it or make it transparent that you address some research questions exploratorily.*

Response: Thank you for your constructive feedback on our research goals paragraph. In the revised manuscript version, we have moved away from the broad labels of 'adaptive' and 'maladaptive' emotion regulation strategies. Instead, we now refer to the specific strategies examined. Additionally, we have revised the paragraph to adopt clearer language overall and to more explicitly indicate which research questions were tested in an exploratory manner. The paragraph on page 6 now reads:

“The present research investigates how trait ER strategy use relates to momentary self-esteem level and variability in adolescence. We addressed two key research questions: First, to what extent do the four ER strategies reappraisal, expressive suppression, social sharing, and reflection explain the level of adolescents’ momentary self-esteem? Second, to what extent do ER strategies explain the variability of adolescents’ momentary self-esteem? Based on trait ER and trait self-esteem associations in adult samples (Gross & John, 2003), we hypothesized that reappraisal is positively, and expressive suppression is negatively associated with momentary self-esteem levels. All remaining associations, that is, those of social sharing and reflection with momentary self-esteem levels, as well as those of all four ER strategies with momentary-self-esteem variability were tested exploratorily.”

R2 #5: Methods. *Can you justify your ESM measure from the literature? Is it a validated measure? Where have you taken it from?*

Response: This is an important remark that was shared with Reviewer 1 (cf., R1 #4). We now clearly reference on page x that the item is adapted from the Rosenberg scale and has been successfully applied in other adolescent samples (Bleckmann et al., 2023).

R2 #6: Methods. *What is the r_{SB} coefficient? I don't know that one.*

Response: Thank you for this question. The coefficient denotes the Spearman-Brown correlation, which is an indicator of internal consistency for scales including less than three items. We note that this information was a little unclear in the manuscript and moved the information on the r_{SB} coefficient from the footnote into the main text. We added the following sentence, before we report internal consistency for the first time in the manuscript (page 9):

“To evaluate the internal consistency of three or more item scales, we report McDonald’s omega (ω ; McDonald, 2013), for two item scales, we report the Spearman-Brown coefficient (r_{SB} ; Eisinga et al., 2013).”

R2 #7: Methods. *Perhaps that’s me and I know many other people ignore that too, but from a measurement POV, you report McD’s Omega highlighting a congeneric measurement model, yet you simply average items.*

Response: Indeed, the use of McDonald’s Omega as a measure of internal consistency is subject to ongoing discussion. McNeish (2018) states that the use of Omega (and composite reliability measures in general) is “appropriate when the items from a scale are unit-weighted to form the total scale score but the scale itself is congeneric”. He further recommends the use of Omega over Cronbach’s Alpha, because its assumptions are not as rigid and more likely to be met in most measurement situations. Other researchers have made similar recommendations (e.g., Hayes & Coutts, 2020). While we have the impression that Omega is increasingly becoming the standard measure of internal consistency, we would certainly be open to include other measures, as well.

R2 #8: Methods. *Can you justify your choice of model? I am not familiar with MELMS. Perhaps keep that in mind, when reading my remark: With your data my guess was that you would conduct a DSEM model which incorporate autoregressive components. At the moment you have included robustness checks which are fine, but I guess the autoregressive component of momentary self-esteem might be quite influential. This shouldn't be a critique and I know that DSEM makes it harder to address the variability component of your analyses. I simply got stuck on the question: Would your effects be significant if your model incorporated the autoregressive effects of MSEt-1 on MSEt.*

Response: Thank you for this thoughtful methodological question. Our study exclusively examines how between-person variables (e.g., trait ER strategy use) predict the level and variability of momentary self-esteem of adolescents, where MELMS have been recognized as preferred model choice (McNeish, 2021). As our models include only between-person predictors, accounting for autoregressive components should not alter the estimated associations between ER strategies and interindividual differences in momentary self-esteem levels or variability. We appreciate bringing up this important point and fully agree that

DSEMs would be the appropriate approach for future studies focusing on within-person processes of momentary self-esteem.

R2 #9: *Methods. Perhaps state that you are referring to ICC(1) here which examines the allocation of variance across different levels whereas the ICC(2) you mentioned in the measures section is a measure of reliability*

Response: Thank you for this helpful suggestion. We have revised the sentence to enhance clarity. It now reads (page 12):

“The **ICC(1)** indicated that 54% of the variance in momentary self-esteem was explained by between-person differences, pointing to substantial within-adolescent variability alongside systematic between-person differences in momentary self-esteem.”

R2 #10: *Discussion. I really like your discussion in general and my remarks are only an re-iteration of the points I have made concerning your introduction. I think, a clear reiteration of research questions and hypotheses would be nice.*

Response: Thank you for your suggestion regarding the discussion. After the revisions in our manuscript, it slightly exceeds the recommended length, so we decided against reiterating the research questions and hypotheses at this stage. We hope that the improved clarity of the introduction, thanks in part to your helpful comments, has made both the overall text and the discussion more accessible.

R2 #11: *Discussion. You do a great job in discussion and evaluating your effects, but I think the discussions falls short of a paragraph on the theoretical impact of the research.*

Response: Thank you for this suggestion. We agree and accordingly added the following paragraph on page 21:

Taken together, our results suggest that ER is relevant for momentary self-esteem level and variability in adolescents' daily lives. Furthermore, the differential associations between ER strategies and self-esteem dynamics highlight the value of fostering adaptive ER from a young age onward. Through the lens of sociometer theory (Leary & Baumeister, 2000), our results indicate that ER plays a key role in self-esteem through social belonging, at least during adolescence. Notably, previous research has shown that adolescent ER varies across social contexts—for example, adolescents are less effective at using reappraisal in social contexts, particularly when they are sensitive to rejection (Silvers et al., 2012), and they are more likely to suppress their emotional expressions around peers than around family members (Wylie et al., 2022). Thus, in the context of the broad literature our findings highlight that learning to regulate one's emotions in social situations represents a key developmental task that is well worth mastering.

R2 #12: *I checked output files within the OSF repository. And there are probably multiple copy&paste mistakes in Table 3 of the manuscript. For example:*

- *File 1.2 melsm_mse_suppression.out:*
 - *R-squared: 0.068 in the file; 0.12 in Table:*

- Variability intercept: 0.846 in the file; 0.79 in the Table
- File 1.2 melsm_mse_reappraisal.out
 - R-squared: 0.066 in the file; 0.12 in Table:

I haven't checked all files; but the authors should check all again and highlight differences and whether these differences impact conclusions/discussion

Response: Thank you for raising these specific points. First, when it comes to the R-Squared values reported in our tables, they represent the proportion of explained variance relative to the respective null model. More specifically, the difference between the R-Squared values from the null model and the model in question was computed and then relativized by the explained variance of the null model. In the original manuscript, we reported this only in the method section, but this important fact was missing in the Table notes. We now also explain this more precisely in the note and use ΔR^2 instead of R^2 to clarify that the value denotes the variance explained beyond the null model.

Second, the variability intercept you mention stems from the STDYX Standardization output in Mplus. As is the convention with these models, our manuscript only reports unstandardized estimates. Again, we missed to specify this in the previous version of the paper. To clarify these points, we added two corresponding sentences in all relevant table notes. The note of Table 3 on page 16 in the manuscript, for example, now reads:

*“Note. Results are based on $n = 8349$ observations nested in 408 individuals. MSE = momentary self-esteem. Estimates concerning momentary self-esteem variability are on a logarithmic scale. **Estimates reflect unstandardized model results.** Estimates in bold font have 95% credible intervals not including zero. $\Delta R^2_{\text{Level}}$ and $\Delta R^2_{\text{Variability}}$ represent the proportion of variance that can be explained by all model predictors relative to a null model.”*

References

- Bleckmann, E., Lüdtke, O., Mueller, S., & Wagner, J. (2023). The role of interpersonal perceptions of social inclusion and personality in momentary self-esteem and self-esteem reactivity. *European Journal of Personality, 37*(2), 187–206.
<https://doi.org/10.1177/08902070221080954>
- Hayes, A. F., & Coutts, J. J. (2020). Use Omega Rather than Cronbach's Alpha for Estimating Reliability. But.... *Communication Methods and Measures, 14*(1), 1–24.
<https://doi.org/10.1080/19312458.2020.1718629>
- Lougheed, J. P., & Hollenstein, T. (2012). A Limited Repertoire of Emotion Regulation Strategies is Associated with Internalizing Problems in Adolescence. *Social Development, 21*(4), 704–721. <https://doi.org/10.1111/j.1467-9507.2012.00663.x>
- McNeish, D. (2018). Thanks coefficient alpha, we'll take it from here. *Psychological Methods, 23*(3), 412–433. <https://doi.org/10.1037/met0000144>
- McNeish, D. (2021). Specifying Location-Scale Models for Heterogeneous Variances as Multilevel SEMs. *Organizational Research Methods, 24*(3), 630–653.
<https://doi.org/10.1177/1094428120913083>
- Rosenberg, M. (1965). *Society and the Adolescent Self-Image*. Princeton University Press.
<https://www.jstor.org/stable/j.ctt183pjhh>
- Vachon, H., Viechtbauer, W., Rintala, A., & Myin-Germeys, I. (2019). Compliance and Retention With the Experience Sampling Method Over the Continuum of Severe Mental Disorders: Meta-Analysis and Recommendations. *Journal of Medical Internet Research, 21*(12), e14475. <https://doi.org/10.2196/14475>
- van Roekel, E., Keijsers, L., & Chung, J. M. (2019). A Review of Current Ambulatory Assessment Studies in Adolescent Samples and Practical Recommendations. *Journal of Research on Adolescence, 29*(3), 560–577. <https://doi.org/10.1111/jora.12471>

Wrzus, C., & Neubauer, A. B. (2023). Ecological Momentary Assessment: A Meta-Analysis on Designs, Samples, and Compliance Across Research Fields. *Assessment*, 30(3), 825–846. <https://doi.org/10.1177/10731911211067538>

Reviewer Reports

Reviewer 1

R1 #1: *I appreciate the opportunity to review the revised manuscript. I went to the manuscript and I have carefully reviewed the changes. Overall, I deem the paper to be improved, particularly in clarity. The authors have adequately addressed all of my comments. Personally I would recommend the manuscript for acceptance.*

Response: Thank you for the positive evaluation of the manuscript and for the time and effort invested in reviewing it.

Reviewer 2

R2 #1: *Thank you very much for your thorough revision. I am basically happy for most parts of the revision - however, there is one point where your rebuttal has not convinced me and I think that this is still an issue.*

"As our models include only between-person predictors, accounting for autoregressive components should not alter the estimated associations between ER strategies and interindividual differences in momentary self-esteem levels or variability. "

I think this is simply not true or it ignores the point that I was trying to make. I try better. Your between-level predictors - the scale part is highly dependent on the within part, so this is not a pure between-level issue. Specifically, I am referring to the random residual variance.

Momentary self-esteem fluctuates - and some adolescents fluctuate more strongly around their mean than others. In the within part, you partial out the trajectory proportion within your data, but the time predictor is insignificant on average. So there is not much to partial out. Within your model you treat these small and strong fluctuations of adolescents as kind of trait variable - meaning that it is assumed that we have an Adolescent A who, across days, fluctuates very strongly in his/her MSE around their personal mean while Adolescent B who is very stable in his/her MSE. So Person A is the unstable MSE person, while person B is the stable one.

The model you propose basically ignores the fact that there might be reasons that are inherent to the situation where MSE was captured that make their MSE fluctuate more/ less strongly. Predictors on the within-level will reduce the residual variance, so your between level correlations will be affected by within-part. Don't get me wrong - I see your point and agree with the fact that there are adolescents in your sample that will be more or less stable in their MSE regardless of the situation, but currently situation specific predictors are ignored which leaves much more variance proportions to potentially correlate with other variables on the between level. The autoregressive example is just one example among others.

Response: Thank you for the positive evaluation of our revision and for clarifying your concern. We fully agree that situational factors, and how individuals perceive and respond to them, play an important role in self-esteem dynamics, which we refer to as self-esteem reactivity. While we have already conducted initial research on self-esteem reactivity, in the present manuscript we decided to take a step back and examine whether variance in MSE dynamics can be explained by ER at the between-person level. Our results provide evidence that ER is associated with self-esteem dynamics, and we agree that examining this association at the within-person level, and investigating how situational and personal characteristics relate to it, will be a very important next step. We see the present work as a starting point for such research. To highlight the importance of this further work, on page 19/20, we added the following part to our outlook chapter in our discussion:

“While our results suggest an association between ER and between-person differences in self-esteem variability, our analyses do not provide insights into the underlying

mechanisms that explain *why* some adolescents vary more in their self-esteem than others. It is possible that adolescents' ER is linked to how randomly their self-esteem fluctuates. Alternatively, ER may be involved in their self-esteem reactivity to specific situations (Wagner et al., 2024) or their exposure to situations that trigger frequent self-esteem changes. Therefore, incorporating situational predictors represents the essential next step to disentangle the contributions of individual and situational factors to self-esteem variability.”

R2 #2: *Additionally, in your discussion you write about robust predictors of variability. In a statistical sense, I of course agree with that conclusion, but from a practical point of view we are talking about borderline-significance. Both significant effects nearly touch the zero in the CIs. Paired with the small regression coefficient and the small explained variance, I think these results should be expressed more reasonably in terms of their significance.*

Response: Thank you for this comment. As self-esteem is a broad construct influenced by many factors, it is inherently difficult to explain large proportions of its variance. While, we agree that our estimates are not large, it should be considered that the estimates for momentary self-esteem variability are on a logarithmic scale (McNeish, 2021). For a better interpretation, model estimates corresponding to momentary self-esteem variability should be transformed back by exponentiating the estimates. For example, a variability estimate of -0.05 for reappraisal on the logarithmic scale corresponds to an estimate of $\exp(-0.05) = 0.95$ on the scale of the outcome. We added a clarification on the estimates corresponding to momentary self-esteem variability in the Results chapter on page 12:

“It should be noted that the estimates for momentary self-esteem variability are reported on a logarithmic scale. To interpret these estimates meaningfully, they should be converted back to the original scale by exponentiating the values. For example, a variability estimate of -0.05 for reappraisal corresponds to $\exp(-0.05) = 0.95$ on the outcome scale (McNeish, 2021).“

At the same time, we have adjusted the tone in some parts in our results and discussion chapters, in line with your and the editor’s suggestions. For example, on page 15 in the discussion we added the following sentence:

“Overall, the effects were comparatively larger for momentary self-esteem levels, as reflected in the higher proportions of explained variance compared to variability.“

References

- McNeish, D. (2021). Specifying Location-Scale Models for Heterogeneous Variances as Multilevel SEMs. *Organizational Research Methods*, 24(3), 630–653.
<https://doi.org/10.1177/1094428120913083>